# Directed self-assembly of fluorescence responsive nanoparticles and their use for real-time surface and cellular imaging

Shane Cheung[1] & Donal F. O'Shea [1]

Directed self-assemblies in water are known as the most efficient means of forming complex higher ordered structures in nature. Here we show a straightforward and robust method for particle assembly which utilises the amphiphilic tri-block co-polymer poloxamer-188 and a hydrophobic fluorophore as the two designer components, which have a built-in ability to convey spatial and temporal information about their surroundings to an observer. Templating of particle self-assembly is attributed to interactions between the fluorophore and hydrophobic segment of the poloxamer. Particle fluorescence in water is quenched but can be induced to selectively switch on in response to temperature, surface adsorption and cellular uptake. The ability of the particles to dynamically modulate emission intensity can be exploited for selective labelling and real-time imaging of drug crystal surfaces, natural fibres and insulin fibrils, and cellular delivery. As particle solutions are easily prepared, further applications for this water-based NIR-fluorescent paint are anticipated.

---

[1] Department of Pharmaceutical and Medicinal Chemistry, Royal College of Surgeons in Ireland, 123 St Stephen's Green, Dublin 2, Ireland. Correspondence and requests for materials should be addressed to D.F.O'S. (email: donalfoshea@rcsi.ie)

The production of nano-sized objects is most efficiently achieved if their formation is driven by a thermo-dynamically favourable self-assembly of their individual components in water, akin to natural systems[1–3]. A highly desirable advanced feature would be for the nano-construct to have an in-built capacity to elicit a responsive output, providing the user with real-time information about the environment in which it is located and/or if its cargo has been delivered. The formation of nanoparticles by directed self-assembly (DSA) of polymer building blocks in which the directing template is a fluorophore offers the potential for such a stimuli-responsive system.

Poloxamers are non-ionic amphiphilic tri-block co-polymers assembled in a hydrophilic-hydrophobic-hydrophilic block sequence comprised of poly(ethylene oxide) (PEO) acting as hydrophilic blocks and poly(propylene oxide) (PPO) as the hydrophobic block (Fig. 1)[4]. There are over 50 commercially available poloxamers (also known as pluronics), which vary in the molar mass of their constituent blocks. The availability of such a series of related polymers provides a valuable set of building blocks from which key physical and chemical properties can be selected at the outset to tailor a specific designer use or function[5]. As such, they continue to receive substantial attention as drug-delivery systems for medicine, surfactants and coating agents for pharmaceutical formulations, constituents of nanoparticles engineered for drug delivery, and biological response modifiers[6]. Several poloxamer/therapeutic combinations are in advanced stage clinical trials, such as the tri-block co-polymer delivery of the chemotherapeutic doxorubicin for treatment of advanced oesophageal adenocarcinoma[7]. The ability of certain poloxamers to re-sensitise multidrug resistant cancer cells is of great medicinal potential, with chemotherapeutics such as doxorubicin showing significant increases in cytotoxicity when a poloxamer delivery mechanism is used[8]. Their surface adsorption properties make them widely used as coating agents for pharmaceutical formulation of drug nanocrystals, synthetic nanoparticles and other industrial material applications[9].

Our chosen fluorophore template was from the NIR-AZA class as they have excellent photophysical characteristics such as tuneable emission maxima between 675 and 800 nm, exceptional photostability and high quantum yields (Fig. 1)[10]. These properties have made them attractive candidates for both in vitro live-cell imaging and in vivo imaging, with potential for clinical applications in fluorescence-guided surgery as they have the optimal wavelengths for minimal light-induced toxicity and maximum penetration of light through body tissue[11–13].

An effective off to on fluorescence switch depends not only on the ability to produce a bright fluorescence signal in response to the desired stimuli, but also on having negligible fluorescence to begin with. When both a dark off state and bright on state are achieved, this provides the greatest contrast between the region of interest and background, improving resolution and permitting real-time continuous image acquisition[14]. Several mechanisms are known for fluorophores, including the NIR-AZA class, to reversibly establish a non-fluorescent state such as photoinduced electron transfer (PeT), internal charge transfer (ICT), molecular aggregation, or solvent quenching[15–18]. Our goal was to develop an off to on responsive NIR-AZA-doped poloxamer nanoparticle, which could self-assemble in water, making it suitable for both bio- and material-imaging in aqueous media. It would be a ground-breaking step forward if these advantageous NIR photophysical properties could be harnessed for off to on responsive imaging, in water, without the need for multi-step synthetic elaboration of the fluorophore to introduce PeT or ICT responsive moieties and water solubilizing groups.

At the outset of our design, poloxamer-188 ($P_{188}$) (commercially known as pluronic F-68 with molecular weight of 8400 Da)

**Fig. 1** Component classes selected for directed self-assembly. General structures of poloxamer tri-block co-polymers and $BF_2$-chelated azadipyrromethene (NIR-AZA) fluorophores

was selected for investigation as it has a high PEO to PPO ratio of ~11:2, and hydrophilic–lipophilic balance (HLB) of 29, making it one of the most hydrophilic poloxamers available[19–21]. It is widely used as an additive to mammalian cell cultures (between 0.1 and 5.0 g/L) to prevent hydrodynamic damage of cells in bioreactors, where its protective function is associated with changing cell membrane fluidity and reducing cell–bubble associations[22]. As a therapeutic, it has completed phase II clinical trials for the treatment of sickle cell disease[23]. When used as a surface coating for nanocrystals of hydrophobic anti-inflammatory drugs, such as naproxen and ibuprofen, significant enhancement in blood circulation times were obtained and increased accumulation in tumour tissue was achieved with the chemotherapeutic paclitaxel[24–26]. The therapeutic ability of $P_{188}$ to reseal plasma membranes damaged due to heat shock or exposure to high-radiation doses has been reported, with computational models illustrating plausible mechanisms by which it can effect cell membrane repair[27, 28]. In this report we show, the DSA of poloxamer/fluorophore nanoparticles in water with a hydrophobic NIR-AZA fluorophore acting as the template for particle assembly. The nanoparticles formed have remarkable off to on fluorescence switching in response to stimuli such as temperature, surface adsorption and cell delivery.

## Results

**Synthesis and characteristics of DSA nanoparticles.** NIR-AZA fluorophore **1**, being entirely water insoluble, was chosen as the self-assembly template for this study as it was anticipated that it could be induced to participate in hydrophobic interactions (Fig. 2a). In organic solvents, such as chloroform, **1** has emission $\lambda_{max}$ at 716 nm and high quantum yield of 0.36, respectively[29]. The DSA using **1** as template and $P_{188}$ as building blocks was achieved by first making a THF solution of **1** and $P_{188}$, following which the THF was removed under vacuum to give a dark green solid. The solid mixture of $P_{188}$-**1** was dissolved in water to produce a final concentration of 5 µM for **1** and 1.19 mM $P_{188}$, and a 1% w/v ratio of $P_{188}$ to water. It should be noted that at this concentration of $P_{188}$, gel formation will not occur even at elevated temperatures[19, 20]. The solution was filtered through a 0.22 µm filter, with no loss of material observed on the filter or by UV–Vis analysis, showing that stable aqueous solutions had been formed (Supplementary Fig. 1).

To exemplify the unique properties of the $P_{188}$/**1** combination, another amphipathic non-ionic polymer, polysorbate 20 ($PS_{20}$), was chosen to make aqueous solutions of **1**, thereby allowing a direct comparison with $P_{188}$. $PS_{20}$ is a fatty acid ester of PEO substituted sorbitan with molecular weight of 1228 Da (Fig. 2a). It is one of the most commonly used non-ionic surfactants in the biotechnology industry and is currently used in formulation of protein biopharmaceuticals[30]. Although both $P_{188}$ and $PS_{20}$

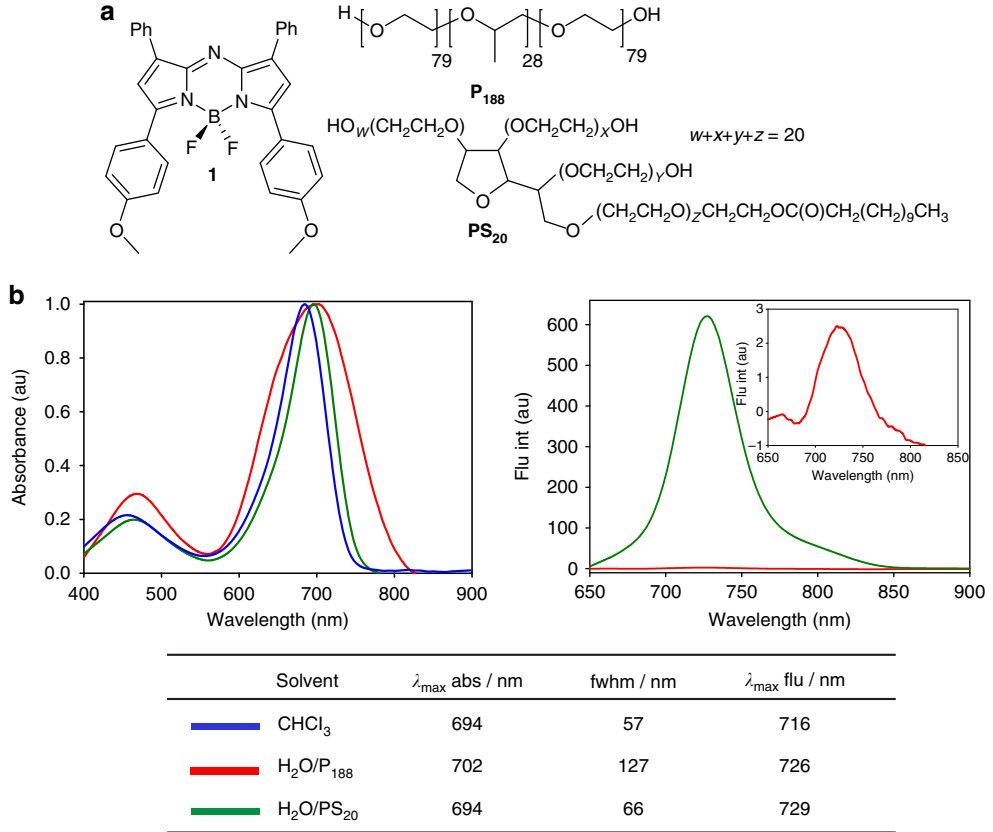

**Fig. 2** Spectral evidence of a DSA using a hydrophobic fluorophore and $P_{188}$. **a** Structures of NIR-AZA **1**, $P_{188}$ and $PS_{20}$. **b** Absorbance spectra, emission spectra and photophysical parameters of **1** in $CHCl_3$ (blue), $H_2O/P_{188}$ (red) and $H_2O/PS_{20}$ (green). 5 µM **1**; 1.19 mM $P_{188}$; 8.14 mM $PS_{20}$

| | Solvent | $\lambda_{max}$ abs / nm | fwhm / nm | $\lambda_{max}$ flu / nm |
|---|---|---|---|---|
| (blue) | $CHCl_3$ | 694 | 57 | 716 |
| (red) | $H_2O/P_{188}$ | 702 | 127 | 726 |
| (green) | $H_2O/PS_{20}$ | 694 | 66 | 729 |

comprise mostly of PEO units with smaller hydrophobic components, only $P_{188}$ has its hydrophilic and hydrophobic components structurally organised in a tri-block manner. $PS_{20}$ was chosen as a comparative polymer system to $P_{188}$ because of this structural difference. Aqueous solution of $PS_{20}$ (1% w/v, 8.14 mM) containing 5 µM **1** was prepared in the same manner as described above for $P_{188}$. Photophysical properties of three different solutions of **1** were first examined. Organic soluble **1** showed an absorbance $\lambda_{max}$ at 694 nm in $CHCl_3$ (Fig. 2b). The absorbance max of aqueous **1**-$P_{188}$ was found to be comparable at 698 nm, but the band was broadened with an increased full width at half maximum (fwhm) value of 127 nm relative to 57 nm in $CHCl_3$. Examination of the properties of **1**-$PS_{20}$ again showed a similar $\lambda$ max value (694 nm), though with narrow band widths of 66 nm and more comparable to the $CHCl_3$ solution.

Examination of emission wavelengths for **1**-$P_{188}$ and **1**-$PS_{20}$ showed them to be comparable at 726 and 729 nm, respectively. Yet, pronounced differences in fluorescence intensities for $P_{188}$ and $PS_{20}$ aqueous solutions were seen, in that emission from **1**-$P_{188}$ was extremely weak in contrast to the notably stronger **1**-$PS_{20}$ (Fig. 2b). When compared directly, the fluorescence intensity as determined by peak area of **1**-$P_{188}$ was 410 times lower than **1**-$PS_{20}$. This variation in emission intensities, together with broadening of the absorption fwhm for the $P_{188}$ solution, affirms that subtle differences must exist between the two aqueous systems. The strongly fluorescent aqueous $PS_{20}$ solution of **1** illustrates that the nature of the microenvironment provided by the polymer surrounding the fluorophore can significantly influence emission intensities. These results indicate that for the $P_{188}$ solution of **1**, fluorophore–water and/or fluorophore–fluorophore interactions are occurring that quench the emission. Encouragingly, the

negligible fluorescence from aqueous **1**-$P_{188}$ is a desirable feature as it fulfils the criteria for the off position of a fluorescence switch.

To gain insights into the size characteristics of the species generated from the combination of **1** and both polymer systems, dynamic light-scattering (DLS) analysis was carried out. DLS measurements, at 25 °C, of aqueous solutions of $P_{188}$ and $PS_{20}$ at the same concentrations used for the photophysical measurements described above gave diameter size values of 6.47 and 9.70 nm, respectively, consistent with literature values (Fig. 3a)[19, 30]. Our interpretation of these values is that $P_{188}$ exists as a unimer in solution, whereas $PS_{20}$ forms micelles at the concentrations used as it is above CMC and the $PS_{20}$ micelle size value obtained is consistent with the literature[31]. The unimer nature of $P_{188}$ is a result of its tri-block molecular structure containing two relatively long hydrophilic PEO chains and a shorter PPO block resulting in self-solubilising unimers (PEO chains enveloping PPO chain) with little driving force towards micelle formation (Fig. 3b). This is consistent with previous literature showing that rt solutions of $P_{188}$ exist only as unimers at concentrations up to 6.1 mM[19]. This lack of rt PPO block self-association is specific to members of the poloxamer family which have the highest HLB values, and is a consequence of their high PEO to PPO ratio. It is this prevalence of rt unimer species that first attracted us to explore $P_{188}$ as a building block for the DSA of nanoparticles, as the lack of self-association of the hydrophobic PPO components in water could allow for a hydrophobic fluorophore template to promote the self-assembly of fluorophore doped entities.

Next, DLS measurements of the aqueous $P_{188}$ solution of **1** was recorded. The sizing results showed that the presence of **1** caused a marked increase in particle size, with a diameter of 103.2 nm obtained (Fig. 3a). The conversion of unimer $P_{188}$ into

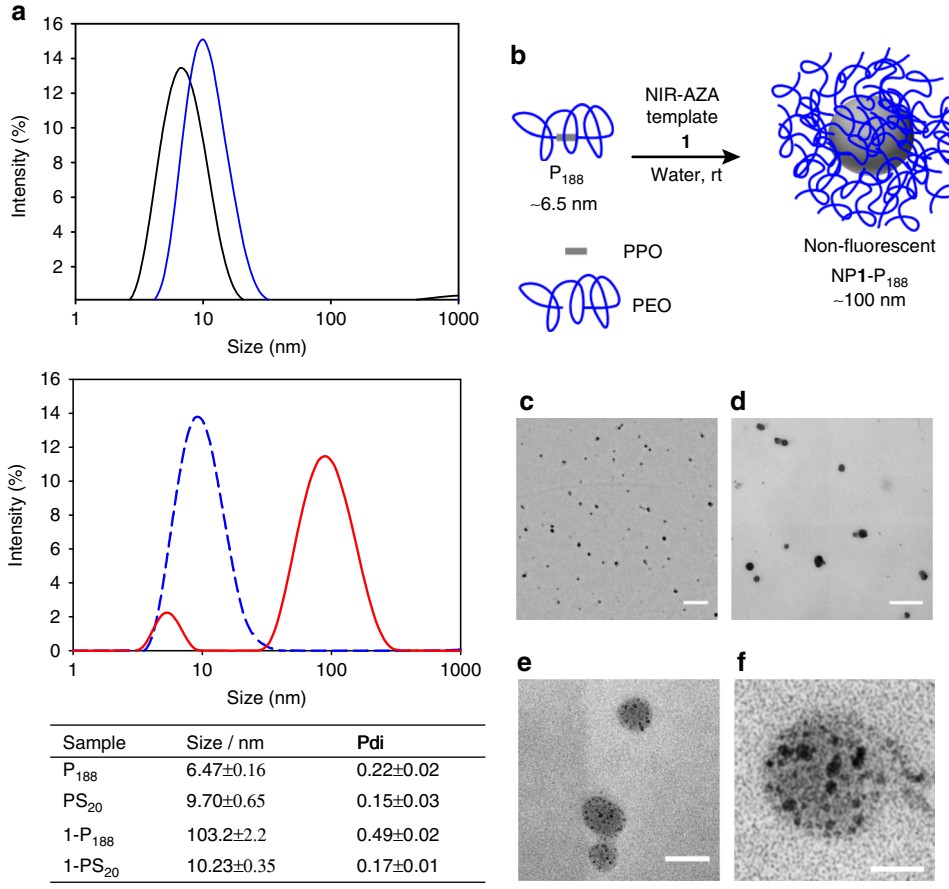

**Fig. 3** Directed self-assembly (DSA) of nanoparticles from $P_{188}$ and **1**. **a** Dynamic light-scattering (DLS) properties of $P_{188}$ and $PS_{20}$ with and without **1** (average of three independent experiments). DLS traces for; $P_{188}$ (solid black); $PS_{20}$ (solid blue), NP**1**-$P_{188}$ (solid red), **1**-$PS_{20}$ (dashed blue). Measurements taken at 25 °C, conc: 5 µM **1**; 1.19 mM $P_{188}$; 8.14 mM $PS_{20}$ (Pdi polydispersity index). **b** Schematic for DSA of **1** doped $P_{188}$ nanoparticles with an outer poly (ethylene oxide) (PEO) (blue) and inner poly(propylene oxide) (PPO) (grey) shells. **c**–**f** Transmission electron microscopy (TEM) images of NP**1**-$P_{188}$ at low and high magnifications with scale bars at 500, 200, 50 and 20 nm respectively

nanoparticle was remarkably clean in that no other nano-sized species were detected except some remaining $P_{188}$ unimer. A plausible rationale for the unimer to nanoparticle conversion is that the hydrophobic attraction between the PPO blocks of $P_{188}$ and **1** template the nanoparticle formation which is thermodynamically more favourable than micelle formation (Fig. 3b)[32,33]. Once formed, the nanoparticles were stable to dilution with no change in particle size following four 1:1 serial dilutions with water (Supplementary Fig. 2). Further evidence that this templated self-assembly is as a result of a unique $P_{188}$ unimer/ fluorophore interaction was obtained by comparison with the DLS measurements for $PS_{20}$ solutions containing **1**. In this case, the $D_h$ increased only marginally by 0.5 nm relative to $PS_{20}$ alone (Fig. 3a). This is consistent with **1** having little impact on $PS_{20}$ micelles and indicates that it is located, as would be expected, within the hydrophobic core. This explanation is also in agreement with the strong fluorescence observed from this solution. Formation of discrete nanoparticles was confirmed using scanning electron microscopy (SEM) and transmission electron microscopy (TEM) imaging. NP**1**-$P_{188}$ solution was placed on a Cu grid and allowed to dry at rt for 16 h, then imaged under vacuum. SEM images showed that individual discrete nanoparticles were present, viewable as spherical or elliptical shaped particles approximately 70 nm in size (Supplementary Fig. 3). This reduction in size of particles from DLS measurements to SEM can be attributed to shrinkage in the dry state, which would be expected to be significant due to the high ratio of

PEO to PPO in $P_{188}$. Higher resolution images with more visible nanoparticle morphology were obtained with TEM imaging which again showed discrete particles (Fig. 3c, d). At higher magnification, the interior particle morphology was notable for the presence of darkened dotted regions of 3–5 nm size spread throughout the particles (Fig. 3e, f). This punctate pattern of dark dots could be attributed to localised areas of higher fluorophore concentration, which have been observed in other poloxamer-based particle systems[34].

As the above results show, stable non-fluorescent particles can be readily produced in water by mixing $P_{188}$ and **1**, exploiting the hydrophobic effect. However, to develop useful responsive outputs from these self-assembled systems, a perturbation to the nanoparticles, and as a consequence to the fluorophore microenvironment, would be required to occur in a selective and predictable manner. As particles comprises $P_{188}$ building blocks, it could be anticipated that interaction of $P_{188}$ units with conditions, surfaces or cells that disrupt or remodel the particle structure may selectively establish off to on fluorescence responses. As such, the fluorescence response of aqueous NP**1**-$P_{188}$ to several differing classes of stimuli including (i) temperature, (ii) drug crystal surfaces, (iii) natural fibres and insulin fibrils and (iv) cellular delivery was investigated.

**Response of NP1-$P_{188}$ to temperature**. We first investigated temperature as a stimulus to illustrate the ability of NP**1**-$P_{188}$ to

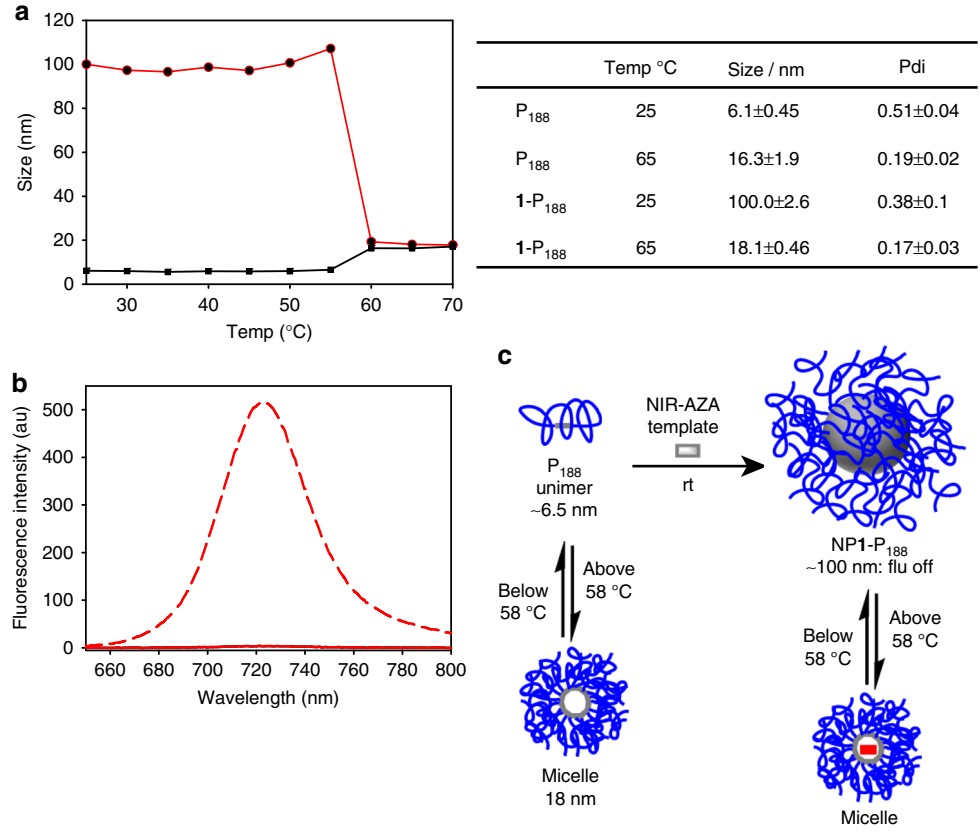

**Fig. 4** Temperature responses of NP**1**-P$_{188}$ indicates involvement of hydration/dehydration in emission switching. **a** Plot of temperature induced size changes and dynamic light-scattering (DLS) data showing the interchange from unimer to micelle for P$_{188}$ (black trace) and from NP**1**-P$_{188}$ to micelle containing **1** (red trace) (average of three independent experiments), Polydispersity index (Pdi). Critical micelle temperatures (CMT) = 58 °C. **b** Temperature induced emission intensity difference for aqueous NP**1**-P$_{188}$. Spectrum taken at 25 °C (red solid trace), 65 °C (red dashed trace). Fluorescence enhancement factor (FEF) determined by peak area. Conc. 5 μM **1**; 1.19 mM P$_{188}$. **c** Schematic summary of temperature induced responses for P$_{188}$ and NP**1**-P$_{188}$

elicit a fluorescence response. Our aim was to gain greater insight into the off to on fluorescence switching mechanism and the inter-relationship between unimer, micelle and nanoparticle states of P$_{188}$ and NP**1**-P$_{188}$. We also aimed to determine the usable temperature range for other sensing and imaging applications by establishing the upper temperature limit for which NP**1**-P$_{188}$ fluorescence remained switched off.

Critical micelle temperature (CMT) is an important parameter for poloxamers as it is only at this temperature that a CMC value for the polymer exists at a given concentration. Increasing temperature to CMT results in a decreased unimer hydration, causing an increased hydrophobicity of block co-polymer molecules and thereby favouring micelle formation. A CMT value of 42 °C has been previously reported for P$_{188}$ at 6.1 mM[19]. As poloxamer CMT values are known to be concentration dependent, this value was measured at 1.19 mM P$_{188}$ (concentration used in previous experiments) by taking DLS particle sizing measurements between 25 and 70 °C at 5° intervals[35]. At temperatures below 55 °C, the P$_{188}$ solution remained in unimer form within the experimental size range of 5.5–6.5 nm. A clear transition occurred between 55 and 60° with micelle formation indicated by an increase in diameter from 6.1 to 18.3 nm (Fig. 4a). Above this, CMT light-scattering measurements show the presence of micelles alone, and cooling the solutions below 50 °C gave rise to re-formation of unimers. This measured CMT value of 58 °C is in broad agreement with other literature values, which were measured at higher concentrations of P$_{188}$[19, 35].

We were intrigued to investigate whether NP**1**-P$_{188}$ would undergo a similar transition upon heating, as increasing temperature would be expected to cause dehydration of the block co-polymer components of the nanoparticle, which may result in a structural reorganisation of the nanoparticle. Experimentally, a solution of NP**1**-P$_{188}$ was heated from 25 to 70 °C, with monitoring of particle size using DLS. Little change was observed until temperature reached 55 °C, above which there was a sharp conversion in particle size from 100 nm to 18.1 nm which could be attributed to a transition from nanoparticle to micelle at the CMT (Fig. 4a). This transition plateaued above 60 °C and heating to higher temperature did not cause further changes. Remarkably, the formation of P$_{188}$ micelles can be induced to occur from two size directions, increasing size up from unimer or decreasing size down from nanoparticle, with both transitions occurring at the same CMT (See Supplementary Fig. 4 for DLS traces).

As the P$_{188}$ micelle core would be more hydrophobic than that of the nanoparticle, the transition from particle to micelle could also result in a change in emission intensity for **1**. This was found to be the case, as in conjunction with the temperature induced structural change from nanoparticle to micelle, a large increase in fluorescence intensity was also observed with a fluorescence enhancement factor (FEF) of 186 upon going from 25 to 65 °C (Fig. 4b and see Supplementary Fig. 5 for temperature induced absorption changes). The nanoparticle to micelle conversion was reversible, as lowering the temperature to below 55 °C reformed 100 nm particles and quenched the fluorescence. To the best of

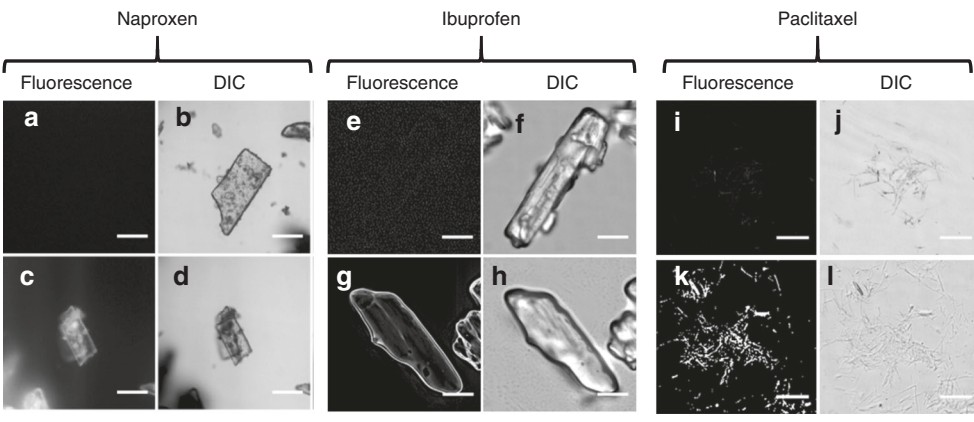

**Fig. 5** Fluorescence (shown in white for clarity) and differential interference contrast (DIC) images of drug crystal following treatment with either $P_{188}$ (control), or NP1P$_{188}$ for 2 h. **a, b** Naproxen following treatment with $P_{188}$. **c, d** Naproxen following treatment with NP1-P$_{188}$. **e, f** Ibuprofen following treatment with $P_{188}$. **g, h** Ibuprofen following treatment with NP1P$_{188}$. **i, j** Paclitaxel following treatment with $P_{188}$. **k, l** Paclitaxel following treatment with NP1P$_{188}$. Scale bars 70 μm. See Supplementary Fig. 6 for alternative views showing images as red fluorescence overlaid on the DIC

our knowledge, this is the first described transition from poloxamer nanoparticle to micelle, and clearly demonstrates how controllable micro-environmental changes around **1** can act as a fluorescence switch.

Taken together, the DLS and emission data clearly illustrate the temperature dependent conversion of NP1-P$_{188}$ to its corresponding micelle, with a resulting switching on of the fluorescence within the micelle as its core is more lipophilic (Fig. 4c). This understanding of the mechanism by which off to on fluorescence response can be induced, in addition to the known surface adsorption properties of $P_{188}$, opens possibilities for the development of new approaches to real-time surface and cellular imaging.

**Response of NP1-P$_{188}$ to hydrophobic drug crystal surfaces**. As the self-assembled fluorophore doped nano-capsules NP1-P$_{188}$ can act as an aqueous reservoir of fluorophore in a non-fluorescent state, the potential exists to selectively induce an emission in response to specific stimuli. It is known that the hydrophobic PPO portion of $P_{188}$ (and other poloxamers) strongly adsorb to hydrophobic surfaces, with the PEO blocks extending, in a brush like manner, towards the aqueous bulk phase providing steric stabilisation[36–39]. Our goal was to exploit this adsorption as an event to elicit a fluorescence response.

It is estimated that around 40% of new drugs have poor solubility in water and 70% of drug candidates fail to proceed to or pass clinical trials due to low solubility and subsequent poor bio-distribution[40]. Established methods for overcoming low solubility and improving bio-distribution involve covalent additions of hydrophilic groups to the drug molecule, formulation with one or more surfactants, or encapsulation in synthetic nanoparticles[41]. Drawbacks of these techniques are the cost, complication and limited scope of modifying the drug molecule, potential side effects of formulating with surfactants and the inherent toxicity of many synthetic nanoparticles[42, 43]. An increasingly viable and important alternative is the use of drug nanocrystals which are dispersed in water as nano-suspensions[44]. This is of specific relevance for the re-investigation of drug candidates, which were previously deemed unsuitable due to low solubility. Drug nanocrystals can also favourably alter the pharmacokinetic profiles of existing drugs and in conjunction with crystal engineering may be utilised during development drug to provide precise crystal forms that are most suitable to the specific clinical delivery challenge[45–48]. The surface coating of nanocrystals with $P_{188}$ has been shown to improve their

therapeutic effect by increasing suspension stability in water with the PEO blocks acting as a steric hindrance to opsonin proteins, allowing them to evade ingestion by macrophages[24–26]. The non-steroidal anti-inflammatory drugs naproxen and ibuprofen and the chemotherapeutic paclitaxel were selected for investigation as representative hydrophobic drugs to gauge whether their crystal surfaces could affect an off to on fluorescence response in water from NP1-P$_{188}$. Each of the drugs was used as a diverse mixture of crystal sizes and morphologies, as supplied by the vendor. Crystal samples of each drug were treated with either $P_{188}$ or NP1-P$_{188}$ for 120 min at rt, following which excess solution was removed. Drug crystal images were acquired using the same microscope settings with 640(14) nm excitation and 705(72) nm emission filters. Promisingly, the $P_{188}$ treated drugs showed no fluorescence (Fig. 5a, e, i), whereas all three drugs treated with NP1-P$_{188}$ clearly showed fluorescence at their crystal surfaces (Fig. 5c, g, k). This was confirmed by comparing and overlaying of the fluorescence images with the corresponding differential interference contrast (DIC) images (Fig. 5b, d, f, h, j, l and Supplementary Fig. 6).

These results illustrate that the process of NP1-P$_{188}$ adsorption onto the crystal surface has re-positioned **1** into a more lipophilic environment thereby achieving a selective surface emission. Real-time dynamic imaging of the process of surface adsorption was next attempted. Experimentally, an aqueous suspensions of drug in a glass bottomed chamber slide was placed on the microscope stage, a selection of crystals focused on and the settings adjusted to ensure minimal background noise. Then an aliquot of NP1-P$_{188}$ was added and images acquired at set time points for 100 s, from which a time-lapse video was produced. All three drugs showed the same response of surface fluorescence emerging within the first 20 s and its intensity continually increasing over the recorded time period (Fig. 6a–c; Supplementary Fig. 7; Supplementary Movies 1–6). Negative control experiments showed that crystals remained non-fluorescent after treatment with $P_{188}$ for 40 min (Supplementary Fig. 8). Plotting the measured change in fluorescence intensity over time for each crystal type showed that a steady increase of intensity occurred over the imaged time period, associated with the adsorption of a mono-layer of $P_{188}$ and **1** on the crystal surfaces (Fig. 6d). Although some variation in the rate of fluorescence increase was noted for the different drug crystal the overall trend was similar for each. Prolonged time-lapse imaging showed that fluorescence continued to increase on the drug surfaces over a 40 min time period (Supplementary Figs. 9, 10; Supplementary Movies 7–12).

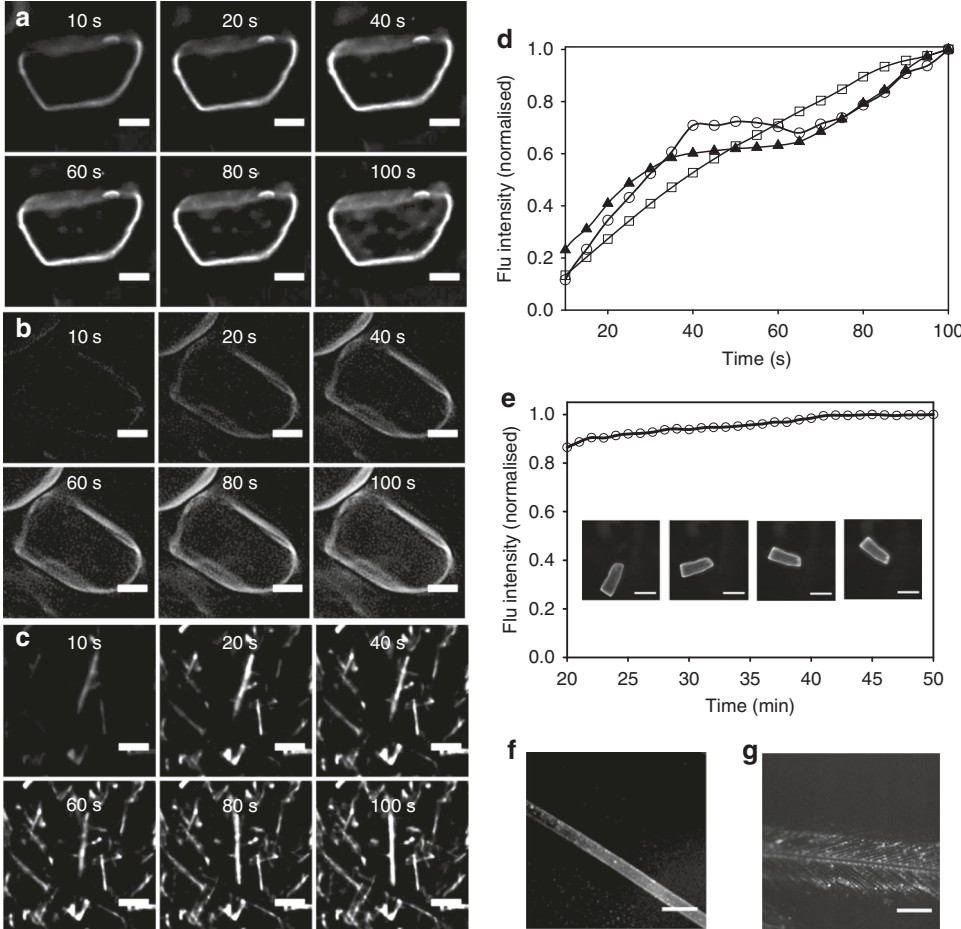

**Fig. 6** Real-time fluorescence imaging of adsorption at drug crystal surfaces over 120 s following treatment with aqueous NP**1**-P$_{188}$ (fluorescence shown in white for clarity) and NP**1**-P$_{188}$ labelling of natural fibre surfaces. **a** Naproxen (Supplementary Movie 1). **b** Ibuprofen (Supplementary Movie 2). **c** Paclitaxel (Supplementary Movie 3). Scale bars = 10 μm. See Supplementary Fig. 7 for alternative views showing images as red fluorescence overlaid on the differential interference contrast (DIC) images and Supplementary Movies 4–6. **d** Plot of increasing drug crystal surface fluorescence over time for the fields of view shown; naproxin (circles), ibuprofen (squares), paclitaxel (triangles). **e** Plot of fluorescence stability of adsorbed **1**/P$_{188}$ on naproxen crystal surface in water over 50 min. Individual crystal shown at 20, 30, 40, 50 min (Supplementary Movie 13). **f** Fluorescence image following treatment of human hair with NP**1**-P$_{188}$, scale bar = 250 μm. **g** Fluorescence images following treatment of feather with NP**1**-P$_{188}$, scale bar = 250 μm. P$_{188}$ negative controls (Supplementary Fig. 11)

The further increase in emission intensity over time could be accounted for as P$_{188}$ and other poloxamers are known to form multi-layer assembles on surfaces[49, 50]. As real-time imaging experiments were conducted in solutions of NP1-P$_{188}$, the stability of the surface fluorescence was investigated by filtering naproxen crystals following a 2 h incubation, twice washing the crystals with water and re-suspended them in water. Following this treatment, real-time imaging was carried out following an individual crystal for 50 min which showed no loss of surface fluorescence (Fig. 6e; Supplementary Movie 13). This approach to real-time imaging has several unique advantages including its simplicity to perform, potential to image and measure rates of adsorption at surfaces and the NIR-fluorescent labelling of hydrophobic surfaces for potential tracking in biological systems.

**Response of NP1-P$_{188}$ to hydrophobic fibres and insulin fibrils**. Two natural fibres, hair and feathers were selected for testing to further illustrate our ability to label hydrophobic surfaces with NP1-P$_{188}$. Samples of human hair and bird feathers were immersed in aqueous solutions of NP1-P$_{188}$ for 2 h and imaged while in the solution (Fig. 6f, g). Pleasingly, both types of fibres showed fluorescence while the control samples treated with P$_{188}$ alone did not

(Supplementary Fig. 11). A wide range of human diseases such as Alzheimer's, Parkinson's, Huntington's and Creutzfeld–Jacobson disease are caused by the inability of cells to correctly fold and transport proteins. This leads to protein mis-folding and the conversion of proteins from their normal water soluble state to hydrophobic aggregates called amyloidal fibrils[51, 52]. Insulin fibrils were used as a model system for initial testing and were formed by heating a pH 1.6 solution of insulin at 60 °C for 16 h with TEM imaging verifying fibril formation (Fig. 7a). Aqueous NP1-P$_{188}$ solution was added to the fibrils for 2 h followed by image acquisition. Lower magnification images of NP1-P$_{188}$ treated insulin fibrils showed distinct regions of fluorescence within the sample (Fig. 7b). At higher magnification, it was possible to discern fibril like thin lines of fluorescence, which were consistent with the fibrils observed in the corresponding DIC image (Fig. 7c, d). In contrast, a P$_{188}$-treated control sample showed no fluorescence when imaged with identical microscope settings, despite fibrils being clearly observable in the DIC image (Fig. 7e, f).

**Response of NP1-P$_{188}$ to cellular delivery**. The use of bio-compatible nano-vessels for drug delivery is a highly attractive prospect. It becomes even more attractive if the carrier vessel can

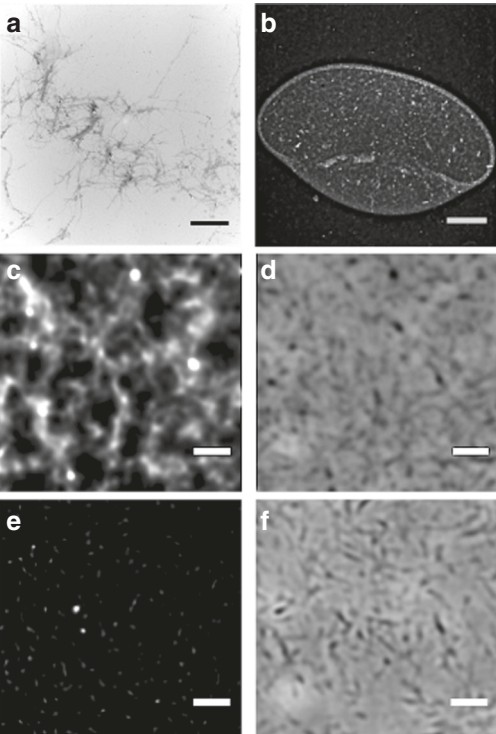

**Fig. 7** NP**1**-P$_{188}$ staining of insulin fibrils (fluorescence shown in white for clarity). **a** TEM images of insulin fibrils (scale bar = 2 μm). **b** Fluorescence imaging of insulin fibrils in water droplet after treatment with NP**1**-P$_{188}$ (scale bar 20 μm). **c** Fluorescence and **d** differential interference contrast (DIC) images of fibrils after treatment with NP**1**-P$_{188}$ (scale bar 5 μm). **e** Fluorescence and **f** DIC images of fibrils after treatment with P$_{188}$ (scale bar 5 μm). See Supplementary Fig. 12 for overlay of fluorescence and DIC

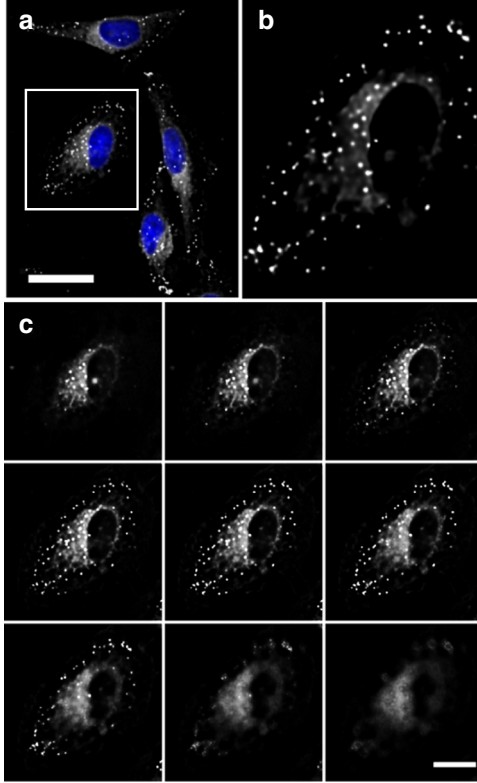

**Fig. 8** Fixed HeLa-Kyoto cell imaging following 2 h incubation with NP**1**-P$_{188}$. **a** Representative multiple cell view. **b** Expansion of single cell area highlighted in **a**. **c** Z-stack analysis of cell shown in **b**. Scale bars = 20 μm. (fluorescence shown in white for clarity). See Supplementary Fig. 14 for an additional independent imaging experiment

self-assemble and provide real-time information about the status of the delivery. Using NP**1**-P$_{188}$ as a representative delivery vehicle for a hydrophobic drug we were uniquely placed to visualise, in real-time, the cellular uptake process. As extracellular NP**1**-P$_{188}$ would be non-fluorescent, it could be anticipated that the particle would release **1** upon cell membrane interaction and internalisation and that this event would establish a fluorescent signal informing an observer that the delivery had occurred. Prior to imaging experiments, the DLS measurement of NP**1**-P$_{188}$ made in cell media gave a particle size of 205 nm, showing that the particles have stability in the media, with the slight increase in size relative to water attributable to changes in the PPO corona (Supplementary Fig. 13). HeLa-Kyoto cells were incubated with NP**1**-P$_{188}$ for 2 h, following which cells were washed, fixed with formaldehyde and nuclei stained with Hoechst 33342 (Fig. 8).

On imaging, we were pleased to observe a strong emission from **1** indicating effective internalisation (Fig. 8a). Images showed that localisation was restricted to the cytosol, with a distinct staining pattern of numerous bright vesicles from the cell periphery to the nucleus and a relatively diffuse perinuclear fluorescence in which vesicles could also be observed (Fig. 8b). Z-stack analysis of individual cells confirmed that all the emissive regions were within the cell cytosol (Fig. 8c). A similar staining pattern in CHO cells has been reported for P$_{188}$ covalently labelled, via an ester functional group, with fluorescein following a 1.5 h incubation[53].

On the basis of the results outlined above, it was anticipated that cellular delivery of **1** could be continuously imaged in real-time without the need for washing or manipulating cells. This would give a dynamic record of the NP**1**-P$_{188}$-mediated delivery process of **1** which could be a valuable tool in assessing its

performance as a drug delivery vector. For live-cell imaging experiments, HeLa-Kyoto cells were seeded on to a chamber slide 24 h before imaging and the slide was placed on the microscope surrounded by an incubator which maintained a constant temperature of 37 °C and 5% CO$_2$. Once cells were in focus, NP**1**-P$_{188}$ was added and images continually recorded at 1 min intervals for 15 min. Following deconvolution, images were compiled as time-lapsed movies with representative still images at 1, 5, 10 and 15 min shown in Fig. 9a, b (Supplementary Movies 14 and 15). The delivery of **1** was observable within minutes following NP**1**-P$_{188}$ treatment, with distinct spot-like regions appearing intracellularly and their number increasing rapidly over time. Using ImageJ analysis software, intracellular fluorescent vesicles were segmented from the image field of view using the threshold function. This allowed the number of vesicles to be counted using the analyse particles function within the software at each time point. Plotting the number of fluorescent vesicles over time showed a sustained increase in fluorescently labelled cellular vesicles, illustrating the ability of NP**1**-P$_{188}$ to effectively report on its delivery of cargo to the cells (Fig. 9c).

Tracking of the motion of individual fluorescent vesicles showed that they followed defined pathways within the cytoplasm indicative of cellular controlled intracellular transport mechanisms (Fig. 9d). An illustration of this can be viewed in Supplementary Movie 16 in which the translocation of an individual vesicle over a 90 s time period is highlighted in blue. In addition to punctate areas of highest fluorescence intensity, a general distribution of fluorescence throughout the cytoplasm was observed to increase over time showing that fluorophore escape from trafficking vesicles was also occurring.

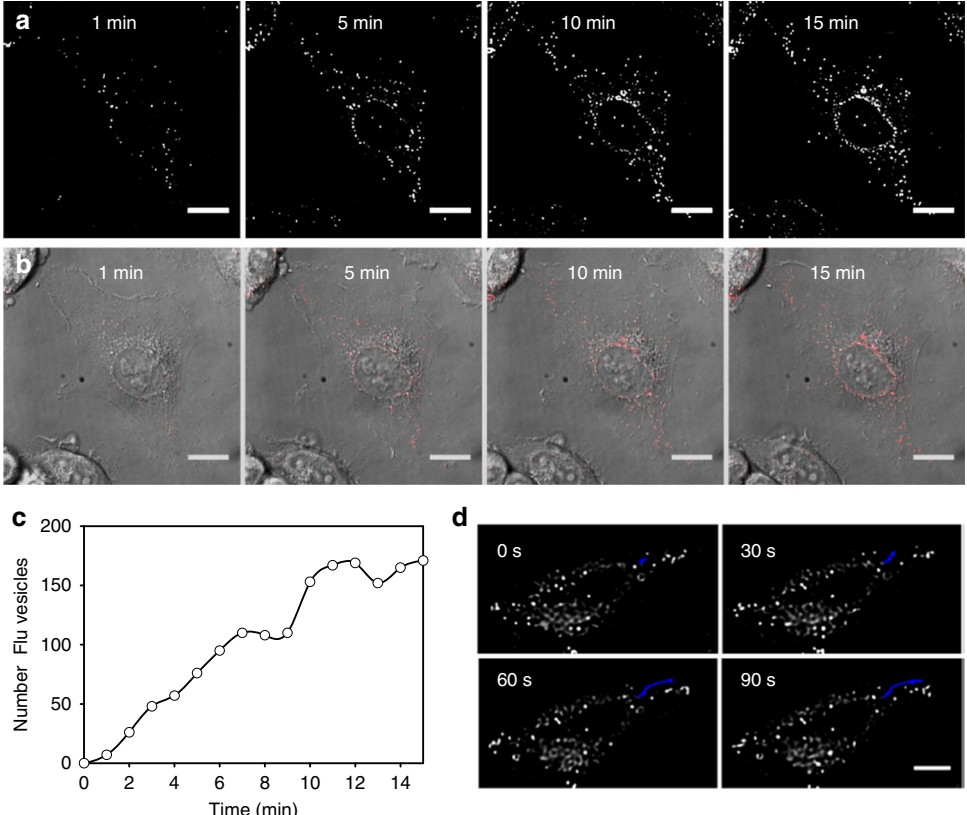

**Fig. 9** Real-time live HeLa-Kyoto cell imaging of uptake, turning on and trafficking of NP**1**-P$_{188}$. **a** NIR-fluorescence (shown in white) from a single cell over 15 min following treatment with NP**1**-P$_{188}$ (Supplementary Movie 14). **b** Overlay of NIR-fluorescence (shown in red) and DIC images from a single cell over 15 min following treatment with NP**1**-P$_{188}$ (Supplementary Movie 15). Scale bars = 20 μm. See Supplementary Fig. 15 and Supplementary Movies 17, 18 for a larger field of view of this experiment. See Supplementary Fig. 16 and Supplementary Movies 19, 20 for images from an additional independent experiments. **c** Plot of increasing number of intracellular fluorescent vesicles over time following treatment of HeLa-Kyoto cells with NP**1**-P$_{188}$. Data acquired from the field of view shown in Fig. 9a. **d** Translocation of a single fluorescent vesicle in a HeLa-Kyoto cell over a 90 s time period (20 min after addition of NP**1**-P$_{188}$). Track of vesicle shown in blue. Scale bar = 20 μm

## Discussion

Encapsulation of water insoluble templates within pre-synthesised molecular capsules or containers is a fundamental research topic that successfully exploits the hydrophobic effect. It has been shown that specifically designed vessels such as deep-cavity cavitands and velcrands are capable of internalising and binding substrates in water. Once bound within a container, communication between the internalised template and the external bulk solution has been achieved by energy or electron transfers, and spin–spin exchanges[54].

In this work, we have devised a two component system that orchestrates a DSA of fluorophore containing nano-capsules, which can self-report on dynamic changes within their local environment. This straightforward and robust method to form discrete nanoparticles in water utilises the tri-block co-polymer P$_{188}$ and a hydrophobic NIR-AZA fluorophore as the only two components. Templating of the self-assembly is attributed to hydrophobic interactions between the fluorophore and PPO segment of the poloxamer leading to nanoparticles of ~100 nm in size. Absorption and emission properties of aqueous nanoparticle solutions showed that the fluorophore was solubilised but non-emissive. Fluorescence quenching was shown to be dependent upon the aqueous hydration of the particles, as heating solutions to their CMT of ~58 °C established a strong fluorescence signal, with DLS measurements showing a structural reorganisation also taking place at this temperature in which nanoparticles were converted to 18 nm sized micelles. This subtle lipophilicity change

in fluorophore microenvironment with resulting off to on fluorescence switching allowed for the demonstration of numerous imaging applications. The known strong affinity of P$_{188}$ through its PPO block for hydrophobic surfaces was exploited for drug crystal and fibre imaging in water. Selective identification and imaging of hydrophobic surfaces was possible as in aqueous solution the bulk NP-P$_{188}$ bound fluorophore is non-fluorescent, yet upon surface adsorption the fluorescence switches on. This was utilised for real-time imaging and semi-quantification of P$_{188}$ adsorption at the crystal surface of drugs such as naproxen, ibuprofen and paclitaxel. As the wavelengths of emission are suitable for in vivo imaging, future uses could include tracking of labelled nanocrystals in vivo. Natural fibres such as hair and feathers also became surface fluorescent upon treatment with NP**1**P$_{188}$ solutions, as did hydrophobic insulin fibrils. As the principles of poloxamer adsorption apply to many hydrophobic surfaces, this fluorescence labelling strategy could have uses ranging from biomedical diagnostics to dye coating applications in the material sciences. Use of the off to on dynamic switching allowed real-time imaging of the cellular uptake of **1**, which showed fluorophore delivery was fast and efficient with selective emission observed predominately from intracellular vesicles. These imaging results point towards a potential future use as a means of tracking drug payload delivery co-packaged in P$_{188}$ nanoparticles.

To conclude, the opportunities offered by our water-based NIR-fluorescent paint for labelling of both biological and material

surfaces are vast and are currently being further explored. In addition, the combined theranostic use of cell uptake switching on of fluorescence with drug delivery via self-assembled poloxamer nanoparticles containing both fluorophore and drug molecules is also under investigation[55]. The scope and generality of directed self-assemblies using amphiphilic tri-block co-polymers with hydrophobic molecular templates would appear to span well beyond the seminal example shown here. Further investigations are ongoing to establish greater insight into the factors controlling these directed self-assemblies, from which predictive models may be developed.

## Methods

**Materials.** Poloxamer $P_{188}$, paclitaxel, ibuprofen and naproxen and human insulin solution were purchased from Sigma Aldrich. All materials were used as received. A molecular weight of 8400 g/mol for $P_{188}$ was used for all concentration calculations. A molecular weight of 1227 g/mol for $PS_{20}$ was used for all concentration calculations. A concentration of 9.5–11.5 mg/mL (1.63–1.98 mM) for human insulin solution (Sigma Aldrich) was used for all calculations. Eight well glass bottomed slides from Ibidi consisted of eight independent wells on glass with a No. 1.5H (170±5 μm) thickness.

**Aqueous DSA of NP1-$P_{188}$.** Solution of NP1-$P_{188}$ in a round bottomed flask was prepared by dissolving $P_{188}$ (0.1 g, 0.0119 mmol) and **1** ($2.715 \times 10^{-5}$ g) in dry THF (15 mL) followed by 1 min sonication. THF was removed under vacuum to give a coloured solid which was further dried under high vacuum for 1 h. Deionised water (8 mL) was added and the solution filtered through a 0.22 μm filter. The resulting solution was transferred to a 10 mL volumetric flask and made up to the mark with deionised water to give a solution consisting of 1.19 mM $P_{188}$ and 5 μM **1**. Solution was allowed to equilibrate overnight prior to use.

**Aqueous $PS_{20}$ solutions of 1.** Solution of **1**-$PS_{20}$ in a round bottomed flask was prepared by dissolving $PS_{20}$ (0.1 g, 0.0814 mmol) and **1** ($2.715 \times 10^{-5}$ g) in dry THF (15 mL) followed by 1 min sonication. THF was removed under vacuum to give a coloured oil which was further dried under a high vacuum for 1 h. Deionised water (8 mL) was added and the solution filtered through a 0.22 μm filter. The resulting solution was transferred to a 10 mL volumetric flask and made up to the mark with filtered deionised water to give a solution consisting of 8.145 mM $PS_{20}$ and 5 μM **1**. Solution were allowed to equilibrate overnight prior to use.

**Control 1% w/v aqueous solutions of $PS_{20}$ and $P_{188}$.** $P_{188}$ (0.1 g, 0.0119 mmol) or $PS_{20}$ (0.1 g, 0.0814 mmol) were dissolved in filtered deionised water (10 mL) to give final concentrations of 1.19 mM and 8.14 mM respectively. All solutions were filtered through a 0.22 μm filter before performing any DLS measurements, spectroscopic analysis, or imaging experiments.

**Formation of amyloidal insulin fibrils.** Human insulin solution (9.5–11.5 mg/mL) was converted into amyloid fibrils following literature procedure[52]. The insulin solution (1 mL) was diluted to 3.725 mL with 100 mM NaCl ($H_2O$) and the pH adjusted to pH 1.6 using 0.5 M HCl ($H_2O$) and the solution heated at 60 °C for 24 h. After cooling to room temperature, the pH was adjusted to pH 7.2 using 0.5 M NaOH ($H_2O$).

**Photophysical properties.** Stock solution of **1** was diluted in chloroform to give a final concentration of 5 μM for UV–visible and fluorescence spectroscopic measurements. Excitation 630 nm, slit widths 5/5 nm.

**UV–Vis and fluorescence analysis of NP1-$P_{188}$ at 65 °C.** A solution of NP1-$P_{188}$ (prepared as described above) was passed through a 0.22 μm filter and analysed via UV–Vis and fluorescence spectrometers at room temperature. Using a mercury thermometer to monitor the temperature this solution was warmed to 65 °C in a water bath and the analysis was repeated. The solution was allowed to cool back to room temperature, and the process was repeated four more times. Excitation = 630 nm, emission range = 650–900 nm, slit widths = 5/5 nm. FEF value was determined from emission peak areas at 25 and 65 °C.

**DLS particle sizing.** Particle size (z-ave) and polydispersity index (PI) were measured using a Zetasizer NanoZS (Malvern Instrument, Malvern, UK) with a 633 nm wavelength He–Ne laser and scattering angle of 173° through the plastic cuvette. Solvent was water (RI 1.0). Samples were prepared by passing 1.5 mL of the solution through a 0.22 μm filter directly into a plastic cuvette. Measurements were made in triplicate at 25 °C. Size and PI of $P_{188}$ (1.19 mM) and NP1-$P_{188}$ (prepared as described above) from 25 to 70 °C were measured using the Zetasizer heat ramp function. After the initial measurement at 25 °C, the temperature was increased in 5 °C increments from 25 to 70 °C with 120 s equilibration after each temperature

increase. Average particle size from three measurements was plotted against temperature. All measurements were performed in triplicate. Data were analysed using Zetasizer Nano software (version 7.1.1).

**Morphological analysis by electron microscopy.** The size and shape of NP1-$P_{188}$ nanoparticles was determined using scanning electron microscopy (SEM Ultra, Carl Zeiss, Oberkochen, Germany) and transmission electron microscopy (TEM Titan, FEI, Oregon, USA). A solution of NP1-$P_{188}$ ($P_{188}$ 1.19 mM; **1** 0.005 mmol) was diluted to ($P_{188}$ 0.48 mM; **1** 0.002 mmol) in water. Overall, 3 μL of this solution was aliquoted on to a 300-mesh formvar-film coated copper grid and air-dried overnight. The grid was imaged by TEM with an accelerating voltage of 300 kV, and by SEM with an accelerating voltage of 5 kV. Size of nanoparticles was measured in ImageJ using the line drawing tools and measure function. Formation of amyloid insulin fibrils was confirmed by TEM (H-7650, Hitachi, Tokyo, Japan). A total of 3 μL of fibril solution (2.5–3 mg/mL) was aliquoted on to a SiO-coated TEM grid, and air-dried overnight. Images were acquired with an acceleration voltage of 100 kV.

**Fluorescence microscope instrumentation.** Fluorescence and DIC images were acquired on an Olympus IX73 epi-fluorescent microscope fitted with an Andor iXon Ultra 888 EMCCD and controlled by MetaMorph (v7.8). Fluorescence illumination was provided by a Lumencor Spectra X light engine containing a solid state light source. DAPI: excitation filter = 390 (40) nm, emission filter = 435 (48) nm. GFP: excitation filter = 482 (18) nm, emission filter = 528 (38) nm. NIR: excitation filter = 640 (14) nm. Emission filter = 705 (72) nm.

**Imaging adsorption of NP1-$P_{188}$ on dry crystal surfaces.** A small spatula tip of naproxen or ibuprofen or paclitaxel was added to deionised water (1 mL) (passed through a 0.22 μm filter) and sonicated for 1 min to create a crystal suspension. Overall, 100 μL of the crystal solution was added to a glass bottomed chamber slide and the slide was not moved for 5 min to allow crystals to settle on the bottom of the well. Excess water was aspirated from the well to leave a coating of crystals on the slide. Solutions of NP1-$P_{188}$ (100 μL) (prepared as described above) was added to a well, and then removed after 120 min. Crystals were allowed to air dry overnight. A control image was produced to confirm that microscope settings did not produce fluorescent artefacts when the crystals were imaged. A control for each crystal type was produced by adding a solution of $P_{188}$ (100 μL) (1.19 mM) to crystals, aspirating it from the well after 120 min and air drying the crystals overnight. NIR-fluorescence and DIC images were acquired with a 20×/0.75 UPlanSApo objective (Olympus). Excitation filter = 640 (14) nm. Emission filter = 705 (72) nm. Exposure 10 ms, LED 40 % and EM gain 1000×.

**Real-time imaging of NP1-$P_{188}$ adsorption on crystal surfaces.** A total of 100 μL of the crystal solution (prepared as outlined above) was added to an eight well chamber slide. The slide was placed on the microscope stage for 5 min to allow crystals to settle on the bottom of the well. Excess water was removed from the well and a field of view (FOV) containing suitable crystals was selected using DIC. Aliquots of NP1-$P_{188}$ (100 μL) (prepared as described above) was then added to the well and NIR-fluorescence and DIC images were immediately acquired at regular intervals while maintaining focus on the crystals in the chosen field of view. A control image was produced to confirm that microscope settings or crystal motion did not produce fluorescent artefacts when the crystals were imaged. A control for each crystal type was produced by adding a solution of $P_{188}$ (100 μL) (1.19 mM) to crystals when acquiring a time-lapse. Early adsorption from 0 to 100 s was imaged with a time-lapse interval of 5 s, exposure 100 ms, LED 80 % and EM gain 2000×. Long-term adsorption from 0 to 40 min was imaged with a time-lapse interval of 2 min, exposure 20 ms, LED 20 % and EM gain 1000×. Mean intensity in the field of view was measured for each time point, and plotted vs time to demonstrate the increasing fluorescence on the crystal surface. Images were acquired using a 60×/ 1.42 oil PlanApo objective (Olympus). Excitation filter = 640 (14) nm. Emission filter = 705 (72) nm. Exposure 10 ms, LED 40% and EM gain 1000×.

**Imaging adsorbed NP1-$P_{188}$ stability on crystal surfaces.** Naproxen crystals were immersed in NP1-$P_{188}$ (100 μL) (prepared as described above) for 120 min on a glass bottomed chamber slide. The NP1-$P_{188}$-containing solution was removed and replaced with water. NIR-fluorescence and DIC images were then acquired ever 60 s for 30 min using a 60×/1.42 oil PlanApo objective (Olympus) to form a time-lapse video. Excitation filter = 640 (14) nm. Emission filter = 705 (73) nm. Exposure 50 ms, LED 60% and EM gain 1×.

**Imaging adsorption of $P_{188}$, NP1-$P_{188}$ on biological fibres.** Human hair and bird feathers were immersed in solutions of NP1-$P_{188}$ (500 μL) (prepared as described above) or $P_{188}$ (500 μL) (1.19 mM), for 120 min in a six well plate. Control images in $P_{188}$ demonstrated fluorescent artefacts did not occur when imaging these materials. NIR-fluorescence and DIC images were acquired with a 5×/0.16 PlanApoChromat objective (Olympus), whereas the fibres remained in the solution. Excitation filter = 640 (14) nm. Emission filter = 705 (72) nm. Exposure 10 ms, LED 60% and EM gain 1000×.

**Fibril imaging**. Insulin fibril solution (100 μL) was added to a glass bottomed chamber slide and treated with of NP1-P$_{188}$ (100 μL, prepared as described above) or P$_{188}$ (100 μL). After 120 min these solutions were removed from the wells and the fibrils were imaged. Control images in P$_{188}$ demonstrated that fluorescent artefacts did not occur when imaging with these settings. NIR-fluorescence and DIC images were acquired with a 100×/1.40 oil UPlanSApo objective (Olympus). Excitation filter = 640 (14) nm. Emission filter = 705 (72) nm. Exposure 100 ms, LED 80 % and EM gain 1000×.

**Cell culture**. HeLa-Kyoto cells were seeded at a density of $1 \times 10^4$ cells per well on eight well removable chamber slide (Millipore), or glass bottomed chamber slides (Ibidi) and allowed to proliferate for 24 h at 5.0% CO$_2$ and 37 °C. Cells were cultured in DMEM supplemented with 10% foetal bovine serum (FBS), 1% penicillin/streptomyosin and 1% L-glutamate.

**Fixed cell imaging**. Hela-Kyoto cells were seeded at a density of $1 \times 10^4$ cells per well on an eight well removable chamber slide (Millipore) and allowed to proliferate for 24 h at 5.0% CO$_2$ and 37 °C. Cells were cultured in DMEM supplemented with 10% foetal bovine serum (FBS), 1% penicillin/streptomyosin and 1% L-glutamate. Solutions of NP1-P$_{188}$ (prepared as described above, 1.19 mM P$_{188}$, 5 μM **1**) were passed through a 0.22 μm filter, 10 μL was then added to 190 μL of DMEM to give a final concentration of 0.0595 mM P$_{188}$ and 0.25 μM **1**. The media was replaced with this solutions, and the cells were incubated for 120 min. Afterwards cells were washed with pre-warmed PBS (37 °C) and fixed with 4% para-formaldehyde (PFA) in PBS for 5 min. PFA was aspirated from the well and the cells were washed three times with PBS. A coverslip was then mounted on the slide using Vectashield containing DAPI to counterstain nuclei. DIC and fluorescence images were acquired on an Olympus IX73 epi-fluorescent microscope fitted and Andor iXon Ultra 888 EMCCD using a 60×/1.42 oil PlanApo objective (Olympus). Z-stacks with 11 optical slices 1 μm apart were acquired in the NIR and DAPI channels. NIR: Excitation filter = 640 (14) nm. Emission filter = 705 (72) nm. Exposure 50 ms, LED 60% and EM gain 1000×. DAPI: Excitation filter = 390 (40) nm. Emission filter = 435 (48) nm. Exposure 5 ms, LED 60% and EM gain 1000×.

**Live-cell imaging**. HeLa-Kyoto cells were seeded at a density of $1 \times 10^4$ cells per well on a glass bottom chamber slide (Ibidi) and allowed to proliferate for 24 h at 5% CO$_2$ and 37 °C. Cells were cultured in iDMEM supplemented with 10% foetal bovine serum (FBS), 1% penicillin/streptomyosin and 1% L-glutamate. The slide was place on the microscope stage surrounded by an incubator to maintain the temperature at 37 °C and CO$_2$ at 5%. DIC imaging was used to choose a field of view and focus on a group of cells. Cell media was removed from the well and replaced with solutions of NP1-P$_{188}$ in DMEM (prepared as described for fixed imaging). Time-lapse imaging was performed by maintaining focus on cells in the chosen field of view and acquiring images every 60 s for 15 min. Mean intensity in the field of view was measured for each time point, this was plotted vs time to demonstrate the increasing intracellular fluorescence.

**Image processing**. All NIR-fluorescence images were restored using fifty iterations of the CMLE method in Huygens Professional deconvolution software (15.10) to correct for out of focus light. DIC images were corrected by flat-field correction to compensate for uneven illumination. A background image ($i_2$) was created by applying a 50 pixel median filter to the source image ($i_1$). Using the Calculator Plus plugin in ImageJ the Divide function was used, corrected image = $(i_1/i_2) \times (k_1 + k_2)$, where $k_1 = I_{mean}$ of ($i_1$) and $k_2 = 0$. The Unsharp Mask filter (weight = 0.7, pixel = 2.0) in ImageJ was applied to fluorescent fibrils images to enhance the contrast.

**Surface fluorescence change over time due to crystal adsorption**. Adsorption of NP1-P$_{188}$ onto the surface of paclitaxel, naproxen, or ibuprofen crystals over time was calculated by measuring the mean intensity ($I_{mean}$) in the field of view at each time point in a time-lapse video. $I_{mean}$ values were normalised from 0 to 1 by dividing the $I_{mean}$ of an image by the highest $I_{mean}$ in the series.

**Counting fluorescence-labelled intracellular vesicles over time**. The number of fluorescent intracellular vesicles in an image after the addition of NP1-P$_{188}$ to wells containing HeLa-Kyoto cells was measured using ImageJ. Fluorescent vesicles were identified using the default threshold tool to produce a binary image, individual vesicles were segmented using the watershed function, and the number of individual vesicles was counted using the analyse particles function. This process was applied to each time frame, and the number of vesicles was plotted vs time.

**Fluorescence-labelled trafficking**. Using the ImageJ plugin Manual Tracking the trajectory of vesicles 20 min after the addition of NP1-P$_{188}$ to wells containing HeLa-Kyoto cells was tracked by manually selecting the same vesicle in each frame of a time-lapse video, and adding a graphical overlay (blue) to display the vesicle track. The local maximum option for centring correction was selected to aid in determining the *xy* coordinates of the vesicle based on its intensity.

**Data availability**. All other data are available from the authors upon reasonable request.

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

## Acknowledgements

D.F.O. is grateful to the Royal College of Surgeons in Ireland for start-up grant funding.

## Author contributions

D.F.O. conceived the project and designed the research. S.C. performed the experiments. D.F.O. and S.C. wrote the manuscript.

## Additional information

**Competing interests:** A patent application has been filed on the synthesis and uses of poloxamer/fluorophore nanoparticles in which D.F.O. has a financial interest. The remaining author declares no financial interests.

