## [Peer Review File · Nature Communications]

Reviewers' comments:

Reviewer #1 (Remarks to the Author):

There have been a number of reports on templating of particle self-assembly which is attributed to hydrophobic interactions between the active substances and PPO segment of poloxamer. Active substances were anticancer drugs (doxorubicin and docetaxel) and inorganic nanoparticles (gold and iron oxide) for molecular imaging and photodynamic therapy.

The sol-gel transition of poloxamer is well-known phenomenon. Especially, The off to on dynamic switching of fluorescence at around 58 oC is not attractive.

Efficient cellular uptake of poloxamer-based nanoparticles is also well-known phenomenon.

Although authors demonstrated a couple of functionality of DSA nanoparticles, these seems to be the combination of previous works.

Based on these understanding, the originality of this work is not enough for the publication.

Reviewer #2 (Remarks to the Author):

This manuscript by Prof. O'Shea describes the self-organization of block polymers, namely PEO-PPO-PEO, which are important classes of amphiphilic polymers that frequently are used in drug delivery. What is interesting is that this self-organization can be template by organic dyes such as the NIR-AZA 1. The interplay of organization and change of the environment of the dye can turn itself on and off according to its environment. Although not new for fluorescence to turn on in this manner, the combination of the dye and polymer integration does exhibit some interesting properties. For example, the self-assembled nanoparticles of 1 and P188 have little fluorescence, but its fluorescence will turn on if triggered with proper stimulations. The author demonstrated that that temperature induces micelle formation and thus turn on fluorescence. Similarly, if the nanoparticles were delivered to living cells and cells will uptake such particles and turn on fluorescence. Overall, this is cleaver way to integrate dyes with flexible polymers to come up new functions. Therefore, this reviewer suggests publication with minor revision.

The manuscript appears too long with more 20 Figures. One would think that a Journal named Nature Communication should publish "short articles" that are to the point. Thus many detail measurements such as light scattering and routine absorption and emission spectra. Table 1 titled parameters of 1 in aqueous solutions, by the first entry is an organic solvent. The TEM images of the nanoparticles appear to be non-uniform and some are connected. Could the authors explain these phenomena? Light scattering reveals that the nanoparticles are around 100 nm, but the TEM results indicate that the nanoparticles are about 50 nm and even smaller (Figure 8). How to reconcile these two results?

Reviewer #3 (Remarks to the Author):

NCOMMS-17-18358A-Z

The manuscript of O'Shea and Cheung describes the formation of direct self-assembly between a long wavelength emitting fluorophores and poloxamers. These give rise to the formation of nanoparticles (DSA Nps) the emission of which can be modulated, or 'switched on' for instance upon binding to crystal surfaces, used for imaging of fibril networks and in cellular applications. Needless to say this work is very interesting! It's a simple but yet a highly important approach that shows great potentials both in academic research as well as in industrial applications.

The manuscript is very well written, clear and the story develops very nicely; a kind of papers one likes to get to review! The authors have carried out in-depth analysis of their systems. Firstly, they focus on the DSA nanoparticles themselves; their photophysical properties in different solvents, demonstrating that 1-P188 is a unique self-assembly which is highly luminescent in comparison to the 1-PS20 analogue, etc. And this is followed by size analysis of these particles using DLS and TEM; part of this work being carried out over a range of temperatures. Their analysis demonstrates how the DSA by the NIR-AZA fluorophores and P188 controls both size and the structural/morphological output. Their mechanical investigation shows that the P188 unimer can form micelles at elevated temperatures that are non-luminescent, and that the mixing of NIR-AZA to the unimer results in the formation of DSA-NP (or NP10P188) which is also non-emissive. However, heating this system gives rise to strong NIR emission due to entrapped NIR-AZA aggregation within the polymer.

While the above is a very nice contribution to the manuscript, it is what follows that really clinches the importance of this work to the field. The authors move towards demonstrating the application of their systems, for instance, by using fluorescence life-imaging of the coating of crystals (of known drugs) by NP1-P188; this allowing for imaging of each crystal due to surface binding of the DSA Nps. I was truly impressed by that approach as a major challenge currently exists within the pharma industry in demonstrating crystallinity of samples and in the formation of drug delivery vectors' that are luminescent and can be used to follow the uptake and delivery of therapeutics in real time without the need of often complex and un-necessary structural modifications of the drug candidates. This result is of major importance and this application demonstrates the novelty of the work presented and what kind of cross-disciplinary impact it can have. The authors further show applications of their system, by using NP1-P188 in the imaging of natural fibre surfaces, such as insulin fibrils, which is currently done by using small single molecules, and importantly as a cellular delivery vehicle. This latter result is elegantly demonstrating with life-imaging, and the use of Z-stacking experiments.

Overall this is very well written manuscript full of ground-breaking developments that have applications across disciplines. The work has been carried out to very high standard, it is clear and exciting. The authors go into great detail to analyze their systems and prove the application of their discovery. Based on novelty and the high standard of the work herein, I am very pleased to recommend the acceptance of this contribution subject to very minor changes. It is a very elegant work and I had great difficulties finding something wrong with it!

Minor points:

A comment on reproducibility should be included occasionally within the captions of the figures (e.g. Fig. 9)

A comment could be made about the plateauing between 40-70 sec in Figure 14

The contrast of the figures in the supporting info and in the life-imaging is better than that sometimes seen in the printed version so perhaps consider the use colours?

Reviewer 1:

- This reviewer expresses concerns with respect to originality stating that “number of reports on templating of particle self-assembly which is attributed to hydrophobic interactions between the active substances and PPO segment of poloxamer. Active substances were anticancer drugs (doxorubicin and docetaxel) and inorganic nanoparticles (gold and iron oxide) for molecular imaging and photodynamic therapy”

Response:

I would disagree with this claim as while poloxamers have been used in drug delivery (as outlined in our introduction) this is the first report which shows how a specific poloxamer (P₁₈₈) and fluorophore (NIR-AZA) can form, through a directed self-assembly, fluorophore doped nanoparticles with the built-in ability to switch on their fluorescence in a predictable and useful manner. This is entirely different from coating gold nanoparticles or delivery of anticancer agents. In addition, it is entirely novel that our DSA particle can dynamically modulate their emission intensity allowing them to be exploited for selective labelling and real-time imaging of drug crystal surfaces, natural fibres and insulin fibrils, and cellular delivery.

- The reviewer states that “The sol-gel transition of poloxamer is well-known phenomenon. Especially, the off to on dynamic switching of fluorescence at around 58C is not attractive.”

Response:

I would disagree with this assessment as there is no sol-gel transition in our manuscript (as stated in the original manuscript) but what is shown, for the first time, is a transition from 100 nm particles to micelles with a resulting off to on switching of emission. The temperature at which this occurs was not of specific relevance but this study does illustrate the mechanism by which the fluorescence turns on (i.e. dehydration of the PPO polymer block). It was clearly stated in the manuscript (text shown below) that the purpose of this study was mechanistic in nature and not for an application.

“Taken together, the DLS and emission data clearly illustrate the temperature dependent conversion of NP1-P₁₈₈ to its corresponding micelle, with a resulting switching on of the fluorescence within the micelle as its core is more lipophilic (Fig. 4c). This understanding of the mechanism by which *off* to *on* fluorescence response can be induced, in addition to the known surface adsorption properties of P₁₈₈, opens possibilities for the development of new approaches to real-time surface and cellular imaging.”

- The reviewer claims that “Efficient cellular uptake of poloxamer based nanoparticles is also well-known phenomenon”

Response:

While uptake of a limited number of solid nanoparticles coated with poloxamer have been reported (as cited in the manuscript introduction) poloxamer based DSA nanoparticles have not been previously described nor has their use for real-time monitoring as they are up taken by cells. What we have presented in this manuscript is entirely different from previous literature reports.

- The reviewer claims that “although a couple of functionality of DSA nanoparticles, these seem to be combination of previous works”.

Response:

The reviewer is incorrect in this comment as no previous publications exist for poloxamer DSA nanoparticles, nor their use for real-time fluorescence imaging. This is the first manuscript from the authors in which a poloxamer (or any other co-block polymer) has been used so it cannot be a combination of our previous works. I would point out that the reviewer offers no citations to “previous works” to substantiate this comment.

Reviewer 2: This reviewer is recommending publication with minor revisions.

- The reviewer comments that “The manuscript appears too long with more 20 figures”

Response:

The manuscript has been shortened (as shown in revised manuscript with changes tracked) to comply with the journals guidelines in terms of word count and the number of Figures. To

achieve this some text has been removed, some figures have been moved to the supplementary information and some figures have been amalgamated.

- Reviewer points out an error in title for Table 1; “titled parameters of 1 in aqueous solutions, but the first entry is an organic solvent.”

Response:

The legend title has been corrected.

- The reviewer comments that “The TEM images of the nanoparticles appear to be non-uniform and some are connected. Could the authors explain these phenomena? Light scattering reveals that the nanoparticles are around 100 nm, but the TEM results indicate that the nanoparticles are about 50 nm and even smaller (Figure 8). How to reconcile these two results?”

Response:

The difference in sizes was explained in the manuscript due to the fact that the DLS sizing is taken in aqueous solution but the SEM and TEM images are taken of dehydrated particles.

The manuscript text which explains this states that “This reduction in size of particles from DLS measurements to SEM can be attributed to shrinkage in the dry state, which would be expected to be significant due to the high ratio of PEO to PPO in P₁₈₈.”

Reviewer 3. This reviewer is recommending publication subject to very minor revisions.

- The reviewer request that a comment on reproducibility should be included occasionally within the captions of the figures.

Response: Additional text has been added to the legends for Fig 3 and 4 stating that the results are an “average of three independent experiments”.

- The reviewer requests that a comment could be made about the plateauing between 40-70 sec in Figure 14

Response: An additional line has been added to the text which states that “While some variation in the rate of fluorescence increase was noted for the different drug crystal the overall trend was similar for each.”

- The reviewer points out that the contrast of the figures in the supporting info and in the life-imaging is better than that sometimes seen in the printed version so perhaps consider the use colours?

Response:

The use of red colour for fluorescence in the paper images has been avoided as per the journal guidelines which requests not to use this colour. As such several duplicates of the images and movies in red have been included in the supplementary information.

I trust that my responses are comprehensive enough to satisfy the reviewers but I remain open to providing further explanations if requested.

REVIEWERS' COMMENTS:

Reviewer #1 (Remarks to the Author):

1. As described in your manuscript (from line 271 to 273), the hydrophobic PPO portion of P188 (and other poloxamers) strongly adsorb to hydrophobic surfaces, with the PEO blocks extending, in a brush like manner, towards the aqueous bulk phase providing steric stabilization. References 36, 37, 38, and 39 supported this interaction between Pluronic and hydrophobic moiety. Every Pluronic-based nanoparticles containing hydrophobic entity were formed through this mechanism. The specific interaction in the formation of DSA nanoparticles also belongs to this interaction. My concern on the originality of your nanoparticle comes from this issue.

2. I could not find the inter-relationship between response of NP1-P188 to temperature and the other phenomena such as response of NP1-P188 to hydrophobic drug crystal surfaces, response of NP1-P188 to hydrophobic fibres and insulin fibrils, and response of NP1-P188 to cellular delivery. Although I totally agree that real time imaging in your manuscript was very useful, I felt that off to on fluorescence switching to temperature change at around 58 °C was not attractive. As I described in the previous review by me, the transition of your nanoparticle can be expected from the sol-gel transition of P 188 at various concentration.

3. However, two reviewers agreed to accept your study with revision, I would not continue to insist my opinion on your manuscript. I hope my review is helpful for your study.

Reviewer #2 (Remarks to the Author):

The authors have addressed this reviewer's issues satisfactorily and the manuscript is now ready to be published in Nature Communications.

Reviewer #3 (Remarks to the Author):

I have reviewed the revised manuscript from O'Shea and co-worker and the response letter to the three reviewers comments. In my own case, the authors have fully answered my suggestions/questions and I am happy with the changes they made in this revised version. As such I am recommending that this version of the manuscript is accepted for publication in Nature Communications as it stands.

Reviewer 1:

1. As described in your manuscript (from line 271 to 273), the hydrophobic PPO portion of P188 (and other poloxamers) strongly adsorb to hydrophobic surfaces, with the PEO blocks extending, in a brush like manner, towards the aqueous bulk phase providing steric stabilization. References 36, 37, 38, and 39 supported this interaction between Pluronic and hydrophobic moiety. Every Pluronic-based nanoparticles containing hydrophobic entity were formed through this mechanism. The specific interaction in the formation of DSA nanoparticles also belongs to this interaction. My concern on the originality of your nanoparticle comes from this issue.

The authors thank the reviewer for this clarification.

2. I could not find the inter-relationship between response of NP1-P188 to temperature and the other phenomena such as response of NP1-P188 to hydrophobic drug crystal surfaces, response of NP1-P188 to hydrophobic fibres and insulin fibrils, and response of NP1-P188 to cellular delivery. Although I totally agree that real time imaging in your manuscript was very useful, I felt that off to on fluorescence switching to temperature change at around 58 °C was not attractive. As I described in the previous review by me, the transition of your nanoparticle can be expected from the sol-gel transition of P 188 at various concentration.

The authors thank the reviewer for this clarification.

3. However, two reviewers agreed to accept your study with revision, I would not continue to insist my opinion on your manuscript. I hope my review is helpful for your study.

Response

The authors thank the reviewer for all their insightful comments.

Reviewer 2:

The authors have addressed this reviewer's issues satisfactorily and the manuscript is now ready to be published in Nature Communications

Response:

The authors thank the reviewer for all their insightful comments.

Reviewer 3.

I have reviewed the revised manuscript from O'Shea and co-worker and the response letter to the three reviewers comments. In my own case, the authors have fully answered my suggestions/questions and I am happy with the changes they made in this revised version. As such I am recommending that this version of the manuscript is accepted for publication in Nature Communications as it stands.

Response:

The authors thank the reviewer for all their insightful comments.